# Do deep nets really need weight decay and dropout?

**Alex Hernández-García & Peter König**
Institute of Cognitive Science
University of Osnabrück, Germany
`{ahernandez,pkoenig}@uos.de`

## Abstract

The impressive success of modern deep neural networks on computer vision tasks has been achieved through models of very large capacity compared to the number of available training examples. This overparameterization is often said to be controlled with the help of different regularization techniques, mainly weight decay and dropout. However, since these techniques reduce the effective capacity of the model, typically even deeper and wider architectures are required to compensate for the reduced capacity. Therefore, there seems to be a *waste* of capacity in this practice. In this paper we build upon recent research that suggests that explicit regularization may not be as important as widely believed and carry out an ablation study that concludes that weight decay and dropout may not be necessary for object recognition if enough data augmentation is introduced.

## 1 Introduction

A recent work by Zhang et al. (2017) suggested that *explicit regularization may improve generalization performance, but is neither necessary nor by itself sufficient for controlling generalization error.* The authors came to this conclusion from the observation that turning off the explicit regularizers of a model does not prevent the model from generalizing—although the performance does become degraded. This contrasts with traditional machine learning involving convex optimization, where regularization is necessary to avoid overfitting and generalize.

Here, we follow up some of the ideas and procedures from (Zhang et al., 2017) to further analyze the need for explicit regularization in convolutional networks. The main difference with their work is that whereas they consider data augmentation one more form of explicit regularization comparable to weight decay (Hanson & Pratt, 1989) and dropout (Srivastava et al., 2014), we argue instead that data augmentation deserves a different classification due to some fundamental properties: Notably, data augmentation does not reduce the effective capacity of the model, can be implemented in a way that the image transformations reflect plausible variations of the real objects, increases the robustness of the model and can be performed on the CPU in parallel to the gradient updates. This difference in the analysis allows us to conclude that weight decay and dropout may not only be unnecessary, but also that their generalization gain can be achieved by data augmentation alone.

## 2 Experimental setup

### 2.1 Data sets and augmentation

We validate our hypotheses on the highly benchmarked data sets ImageNet (Russakovsky et al., 2015) ILSVRC 2012, CIFAR-10 and CIFAR-100 (Krizhevsky & Hinton, 2009). ImageNet consists of about 1.3 M high resolution images that we resize into 150 x 200 pixels and is labeled according to 1,000 object classes; CIFAR-10 and CIFAR-100 consist of 50,000 32 x 32 pixels images, labeled into 10 and 100 classes respectively. In all cases we divide the pixel values by 255 to scale them into the range $[0, 1]$ and use floating precision of 32 bits. So as to analyze the role of data augmentation, we test the network architectures presented above with two different augmentation schemes as well as with no data augmentation at all:

***Light*** **augmentation.** This scheme is adopted from the literature, for example (Goodfellow et al., 2013; Springenberg et al., 2014), and performs only horizontal flips and horizontal and vertical translations of 10% of the image size.

***Heavier*** **augmentation.** This scheme performs a larger range of affine transformations, as well as contrast and brightness adjustment. See the details of these transformations in the Appendix A

The choice of the parameters is arbitrary and the only criterion was that the objects are still recognizable, by visually inspecting a few images. We deliberately avoid designing a particularly successful scheme. In the case of ImageNet we additionally perform a random crop of 128 x 128 pixels, or simply a central crop if no augmentation is added.

## 2.2 NETWORK ACRCHITECTURES

We perform our experiments on two popular architectures that have achieved successful results in object recognition tasks: the all convolutional network, All-CNN (Springenberg et al., 2014) and the wide residual network, WRN (Zagoruyko & Komodakis, 2016). While All-CNN has a relatively small number of layers and parameters, WRN is rather deep and has many more parameters.

**All convolutional net.** All-CNN consists of only convolutional layers with ReLU activations. The CIFAR version has 12 layers and 1.3 M parameters and the Imagenet counterpart has 16 layers and 9.4 M parameters. The architectures can be described as follows:

$$
\begin{array}{c|c}
\text{CIFAR} & \begin{array}{c} 2\times96\text{C}3(1)\text{--}96\text{C}3(2)\text{--}2\times192\text{C}3(1)\text{--}192\text{C}3(2)\text{--}192\text{C}3(1)\text{--}192\text{C}1(1) \\ \text{--}N.Cl.\text{C}1(1)\text{--Gl.Avg.--Softmax} \end{array} \\
\hline
\text{ImageNet} & \begin{array}{c} 96\text{C}11(2)\text{--}96\text{C}1(1)\text{--}96\text{C}3(2)\text{--}256\text{C}5(1)\text{--}256\text{C}1(1)\text{--}256\text{C}3(2) \\ \text{--}384\text{C}3(1)\text{--}384\text{C}1(1)\text{--}384\text{C}3(2)\text{--}1024\text{C}3(1)\text{--}1024\text{C}1(1) \\ \text{--}N.Cl.\text{C}1(1)\text{--Gl.Avg.--Softmax} \end{array}
\end{array}
$$

where $K\text{C}D(S)$ is a $D \times D$ convolutional layer with $K$ channels and stride $S$, followed by batch normalization and a ReLU non-linearity. *N.Cl.* is the number of classes and Gl.Avg. refers to global average pooling. The CIFAR network is identical to the All-CNN-C architecture in the original paper, except for the introduction of the batch normalization layers (Ioffe & Szegedy, 2015). The ImageNet network is identical to the original version, except for the batch normalization layers and a stride of 2 instead of 4 in the first layer to compensate for the reduced input size. See the Appendix B for the hyperparameters details.

**Wide Residual Network.** WRN is a modification of ResNet (He et al., 2016) that achieves better performance with fewer layers, but more units per layer. Although in the original paper several combinations of depth and width are tested, here we choose for our experiments the WRN-28-10 version (28 layers and about 36.5 M parameters), which is reported to achieve the best results on CIFAR. It has the following architecture:

$$16\text{C}3(1)\text{--}4\times160\text{R}\text{--}4\times320\text{R}\text{--}4\times640\text{R}\text{--BN--ReLU--Avg.}(8)\text{--FC--Softmax}$$

where $K\text{R}$ is a residual block with residual function BN–ReLU–$K$C3(1)–BN–ReLU–$K$C3(1). BN is batch normalization, Avg.(8) is spatial average pooling of size 8 and FC is a fully connected layer. The stride of the first convolution within the residual blocks is 1 except in the first block of the series of 4, where it is 2 to subsample the feature maps. See the Appendix B for all the details.

## 2.3 EXPERIMENTS

In order to analyze the need for weight decay and dropout and test the hypothesis that data augmentation alone might provide the same generalization gain, we first train all networks with both regularizers, as in the original papers, and without them. In all cases we also train with the three data augmentation schemes presented above: light, heavier and no augmentation. The test accuracy we report comes from averaging the softmax posteriors over 10 random *light* augmentations, similarly to previous works (Krizhevsky et al., 2012; Simonyan & Zisserman, 2014).

## 3    RESULTS AND DISCUSSION

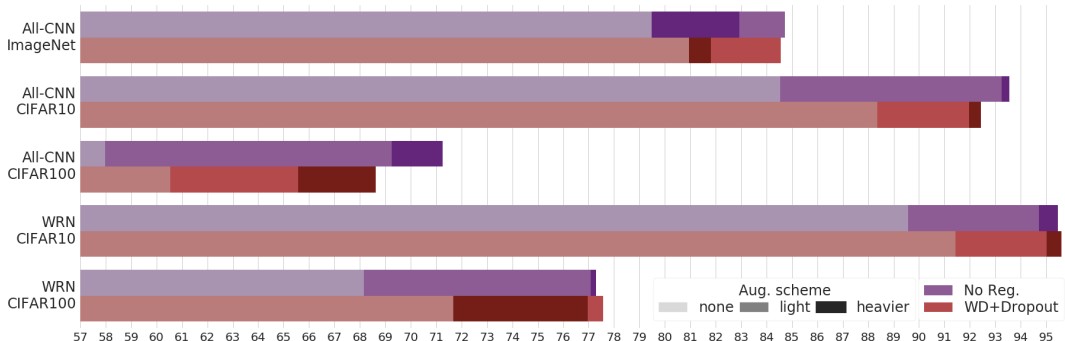

Figure 1: Comparison of the test accuracy of the networks All-CNN and WRN on ImageNet ILSVRC 2012 (top-5), CIFAR-10 and CIFAR-100 trained with weight decay and dropout (red bars) and without them (purple bars). The different shades within each bar correspond to different levels of data augmentation. The results suggest that data augmentation alone can achieve even better performance or only slightly worse than the models trained with both weight decay and dropout.

As expected, weight decay, dropout and data augmentation are all successful in reducing the generalization error. However, some relevant observations can be made from the results shown in Figure 1. Most notably, it seems that data augmentation alone is able to regularize the model as much as in combination with weight decay and dropout and in some cases it clearly achieves better performance, as in the case of All-CNN. In the experiments with WRN on CIFAR-10 and CIFAR-100, slightly better results are obtained with regularization, but the difference is only 0.13 and 0.28 % respectively, which should not be considered significant. Also, note that on CIFAR-100 with heavier augmentation the performance without regularization is indeed slightly higher.

It is important to observe that in all cases we have trained with the hyperparameters reported by the authors in the original papers, highly optimized to achieve state-of-the-art results. However, after removing weight decay and dropout, the value of the hyperparameters, for example the learning rate, is unlikely to be optimal. As a matter of fact, we have observed that without regularization and data augmentation a higher learning rate achieves better performance. Therefore, the fact that the results without regularization reported here are disadvantaged reinforces our hypothesis.

Another piece of evidence in this regard is that in those cases where the regularization hyperparameters were not optimized by the authors, for instance with All-CNN on CIFAR-100, the models without regularization are clearly superior. This suggests that data augmentation can easily adapt and provide high benefits regardless of the hyperparameters, whereas the amount of weight decay and dropout needs to be adjusted according to the architecture, the data set or the amount of training data, for example. This is one of the main reasons why showing that it is possible to achieve equivalent performance without explicit regularization is a relevant result.

## 4    CONCLUSION

In this work we have shed some more light on the need for weight decay and dropout regularization to train convolutional neural networks for object recognition. These techniques have been widely added to most deep neural networks because they clearly provide successful control of overfitting. However, the literature lacks some systematic analysis about the need for explicit regularization when other implicit regularizers such as SGD, convolutional layers, batch normalization or data augmentation are present. We depart from the work by Zhang et al. (2017), where it is suggested that explicit regularization might not be necessary for achieving generalization, and perform a set of ablation studies on several architectures and data sets that show that weight decay and dropout may not only be unnecessary, but also that their benefits can be provided by data augmentation alone.

We leave for future work the analysis of these results on a larger set of architectures and data sets, as well as further exploring the benefits of data augmentation compared to explicit regularization.

ACKNOWLEDGMENTS

This project has received funding from the European Union's Horizon 2020 research and innovation programme under the Marie Sklodowska-Curie grant agreement No 641805.

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

## A    DETAILS OF THE HEAVIER DATA AUGMENTATION SCHEME

- Affine transformations: $\begin{bmatrix} x' \\ y' \\ 1 \end{bmatrix} = \begin{bmatrix} f_h z_x \cos(\theta) & -z_y \sin(\theta + \phi) & t_x \\ z_x \sin(\theta) & z_y \cos(\theta + \phi) & t_y \\ 0 & 0 & 1 \end{bmatrix} \begin{bmatrix} x \\ y \\ 1 \end{bmatrix}$

- Contrast adjustment: $x' = \gamma(x - \overline{x}) + \overline{x}$

- Brightness adjustment: $x' = x + \delta$

Table 1: Description and range of possible values of the parameters used for the heavier augmentation. $B(p)$ denotes a Bernouilli distribution and $\mathcal{U}(a, b)$ a uniform distribution.

| Parameter | Description | Range |
|---|---|---|
| $f_h$ | Horizontal flip | $1 - 2B(0.5)$ |
| $t_x$ | Horizontal translation | $\mathcal{U}(-0.1, 0.1)$ |
| $t_y$ | Vertical translation | $\mathcal{U}(-0.1, 0.1)$ |
| $z_x$ | Horizontal scale | $\mathcal{U}(0.85, 1.15)$ |
| $z_y$ | Vertical scale | $\mathcal{U}(0.85, 1.15)$ |
| $\theta$ | Rotation angle | $\mathcal{U}(-\frac{\pi}{180}22.5, \frac{\pi}{180}22.5)$ |
| $\phi$ | Shear angle | $\mathcal{U}(-0.15, 0.15)$ |
| $\gamma$ | Contrast | $\mathcal{U}(0.5, 1.5)$ |
| $\delta$ | Brightness | $\mathcal{U}(-0.25, 0.25)$ |

## B    HYPERPARAMETERS

We set the same training parameters as in the original paper in the cases they are reported. Both All-CNN networks are trained using stochastic gradient descent, with fixed momentum 0.9 and learning rate of 0.01 and decay factor of 0.1. The batch size for the experiments on ImageNet is 64, we train during 25 epochs decaying the learning rate at epochs 10 and 20. On CIFAR, the batch size is 128 and we train during 350 epochs decaying at epochs 200, 250 and 300. The kernel parameters are initialized according to the Xavier uniform initialization (Glorot & Bengio, 2010). In the case of WRN we use SGD with batch size of 128, during 200 epochs, with fixed Nesterov momentum 0.9 and learning rate of 0.1 multiplied by 0.2 at epochs 60, 120 and 160. The kernel parameters are initialized according to the He normal initialization (He et al., 2015).

All the experiments are performed on the neural networks API Keras (Chollet et al., 2015) on top of TensorFlow and on a single GPU NVIDIA GeForce GTX 1080 Ti.

