# OpenReview forum: "Do deep nets really need weight decay and dropout?"
_ICLR.cc/2018/Workshop — Reject_

### Official Review · AnonReviewer3 · 2018-03-06
**No need for regularization in ConvNets?**

**Rating:** 5
**Confidence:** 3

**Review:**

The paper argues through experiments that weight decay and dropout are not necessary compared with extensive data augmentation.

It is an interesting point to make, and confirms with some of the experience in the literature. But to me this conclusion is quite obvious even without this paper, and the usefulness is narrower than it was presented. In many practical problems that are not about images, it is hard to have such extensive data augmentation.

In the last paragraph in section 2.1, it says "We deliberately avoid designing a particularly successful scheme." Can there be more clarification on this?

---

### Official Review · AnonReviewer2 · 2018-03-09
**Dropout is not widely believed to be important in presence of BN**

**Rating:** 4
**Confidence:** 5

**Review:**

This paper performs an experimental study comparing the benefits of data augmentation to those of weight decay and dropout for image classification using convolutional networks.

The paper is not about a novel idea, but rather about evaluating the utility of certain types of regularization used for neural networks.

I do not think the study has high significance yet, since the results do not lead to novel or important generalizable insights.

Cons:

The two types of "explicit" regularization considered are weight decay and dropout. But the tested networks all use batch normalization (BN), which is rather powerful regularizer. It has been known since the introduction of BN that it reduces or altogether removes the need for other regularization methods such as dropout, and many networks with BN do not use dropout. Clearly, the generalization gains for networks without weight decay and dropout are coming not just through data augmentation alone as claimed, but also from BN. Based on these experiments alone, it also doesn't seem that one can claim that weight decay will not help in general.

---

### Author Response · Authors · 2018-03-07
**Added additional results from training WRN on ImageNet**

We have updated the pre-print manuscript on arXiv ( https://arxiv.org/abs/1802.07042 ) after getting the results of WRN trained on ImageNet, which further support the main hypothesis of the paper.

---

### Decision · Program_Chairs · 2018-03-20
**ICLR 2018 Workshop Acceptance Decision**

**Decision:**

Reject

**Comment:**

Based on the reviews, this paper has not been accepted for presentation at the ICLR workshop. However, the conversation and updates can continue to appear here on OpenReview.